# Unveiling Prevalence, Risk Factors, and Outcomes of Hepatitis D Among Vulnerable Communities in Romania

**DOI:** 10.3390/v17010052

**Published:** 2024-12-31

**Authors:** Liana Gheorghe, Speranta Iacob, Irma Eva Csiki, Mihaela Ghioca, Razvan Iacob, Ileana Constantinescu, Bogdan Chiper, Laura Huiban, Cristina Muzica, Irina Girleanu, Nicoleta Tiuca, Sorina Diaconu, Daniela Larisa Sandulescu, Ion Rogoveanu, Andra Iulia Suceveanu, Florentina Furtunescu, Corina Pop, Anca Trifan

**Affiliations:** 1Fundeni Clinical Institute, 022328 Bucharest, Romania; drlgheorghe@gmail.com (L.G.); iecsiki@gmail.com (I.E.C.); lita.mihaela.corina@gmail.com (M.G.); raziacob@gmail.com (R.I.); ileana.constantinescu@imunogenetica.ro (I.C.); bogdan@iconika.ro (B.C.); 2“Carol Davila” University of Medicine and Pharmacy, 050474 Bucharest, Romania; nicoltiuca@gmail.com (N.T.); sorinadiac@yahoo.com (S.D.); ffurtunescu@gmail.com (F.F.); cora.pop@gmail.com (C.P.); 3Academy of Romanian Scientists (AOSR), 030016 Bucharest, Romania; 4Fundeni Clinical Institute, University of Economic Studies, 70167 Bucharest, Romania; 5Department of Gastroenterology, Grigore T. Popa University of Medicine and Pharmacy, 700115 Iasi, Romania; huiban.laura@yahoo.com (L.H.); lungu.christina@yahoo.com (C.M.); gilda_iri25@yahoo.com (I.G.); ancatrifan@yahoo.com (A.T.); 6St. Spiridon Emergency Hospital, 700111 Iasi, Romania; 7Department of Internal Medicine II and Gastroenterology, University Emergency Hospital, 050098 Bucharest, Romania; 8Department of Gastroenterology, Research Center of Gastroenterology and Hepatology, University of Medicine and Pharmacy, 200349 Craiova, Romania; larisasandulescu@yahoo.com (D.L.S.); ionirogoveanu@gmail.com (I.R.); 9Department of Gastroenterology, Emergency County Hospital, 200642 Craiova, Romania; 10Faculty of Medicine, Ovidius University, 900573 Constanta, Romania; andrasuceveanu@yahoo.com; 11Department of Public Health and Management, National Institute of Public Health, 050463 Bucharest, Romania

**Keywords:** delta hepatitis, vulnerable population, prevalence, advanced fibrosis

## Abstract

Background: Hepatitis B (HBV) and Delta (HDV) virus infections pose critical public health challenges, particularly in Romania, where HDV co-infection is underdiagnosed. Methods: This study investigates the epidemiology, risk factors, and clinical outcomes of HBV/HDV co-infection in vulnerable populations, leveraging data from the LIVE(RO2) program. Conducted between July 2021 and November 2023, the program screened 320,000 individuals across 24 counties, targeting socially disadvantaged groups such as rural residents, the Roma community, and those lacking health insurance. Results: Among 6813 hepatitis B surface antigen (HBsAg)-positive individuals, HDV antibody prevalence was 4.87%, with active replication confirmed in 75.6% of HDV-positive cases. Regional disparities emerged, with higher HDV prevalence and replication rates in the Eastern region compared to the South. HDV-positive individuals were more likely to be younger, male, and from rural or socioeconomically disadvantaged backgrounds. Clinically, HDV co-infection correlated with increased liver stiffness, advanced fibrosis stages, and lower steatosis levels compared to HBV mono-infection. Psychiatric comorbidities were more prevalent among HDV-positive patients, highlighting the need for integrated care. Conclusions: This study underscores the urgent need for targeted public health interventions, including enhanced screening, education, and access to novel antiviral therapies like bulevirtide to address the significant burden of HBV/HDV co-infection in Romania.

## 1. Introduction

Hepatitis B Virus (HBV) and Hepatitis Delta Virus (HDV) infections remain critical public health challenges. In Romania, where HBV prevalence remains significant, HDV co-infection represents a critical yet underdiagnosed public health issue [1,2]. Globally, estimates suggest that 12–72 million people are affected by HDV, with prevalence varying widely by geographic location. Three large meta-analyses estimated the pooled global seroprevalence of HDV infection to be 0.2–1.0% among the general population, 4.5–14.6% among people who are hepatitis B surface antigen (HBsAg)-positive, 14.6–18.6% among patients with chronic liver disease attending hepatology clinics [3,4], and even up to >20–30% in Romanian tertiary gastroenterology clinics [5]. This elevated prevalence underscores Romania’s status as an endemic region for HDV, necessitating urgent public health interventions. The underdiagnosis of HBV and HDV is primarily attributed to insufficient screening programs and limited public awareness, resulting in many patients presenting with advanced liver disease. This delay in diagnosis undermines the opportunity for timely intervention, further exacerbating the burden of severe outcomes such as hepatocellular carcinoma (HCC) and liver-related mortality. HDV co-infection is associated with markedly worse outcomes compared to HBV mono-infection. Meta-analyses have shown that HDV doubles the risk of HCC and significantly increases the likelihood of hepatic decompensation, liver transplantation (LT), and liver-related mortality [6,7]. Factors such as low HBV-DNA levels, elevated ALT levels, and residence in endemic regions—including countries like Romania—are strong predictors of HDV infection [8]. Expanding routine HDV screening in all HBsAg-positive persons is crucial for improving outcomes. Addressing these gaps in diagnosis and consecutive antiviral treatment with bulevirtide could significantly reduce the burden of liver-related morbidity and mortality associated with HDV. The factors contributing to the high HBV/HDV prevalence in Romania are multifaceted. Historical data suggest that the introduction of anti-HBV vaccination programs in other countries has led to a significant decline in HDV prevalence, particularly among younger populations [9,10]. However, Romania has not experienced a similar decline, which may be attributed to socio-economic challenges, healthcare access issues, and insufficient awareness regarding the transmission of these viruses [5]. As such, understanding the epidemiology of HBV and HDV in Romania not only sheds light on local health challenges but also aligns with global trends observed in other endemic regions, emphasizing the need for targeted screening and linkage to care strategies. To address the significant public health challenge posed by HBV and HDV infections, the LIVE(RO2) program was launched in Romania as a comprehensive national screening initiative. Funded through the POCU/308/4/9/ “Increasing the number of people benefiting from health programs and services oriented towards prevention, early detection (screening), diagnosis, and early treatment for major diseases” call, this program prioritized the early detection and management of viral hepatitis. The initiative was strategically designed and financed under the National Strategy on Social Inclusion and Poverty Reduction, acknowledging that impoverished and vulnerable populations bear a disproportionate burden of illness and mortality compared to the general population [11]. Socioeconomic disparities further exacerbate health risks, including the spread of hepatitis virus infections. The primary objectives of the program included identifying individuals with HBV and HDV infections, particularly within vulnerable groups, and ensuring they receive timely access to appropriate medical care and preventive interventions. Additionally, the initiative aimed to gather critical epidemiological and clinical data on the prevalence of HBV and HDV, along with associated risk factors, to guide future public health policies and strategies.

## 2. Materials and Methods

### 2.1. Study Population and Sample Design

The seroprevalence and associated clinical characteristics of HBV/HDV co-infection were investigated through a screening program supported by the European Social Fund (ESF), which established specific criteria for targeting vulnerable populations. The program was conducted in the Southern (POCU/755/4/9/136208) and Eastern (POCU/755/4/9/136209) parts of Romania between July 2021 and November 2023. The targeted participants were adults (aged ≥ 18 years) belonging to vulnerable groups, as defined by the ESF criteria adapted by the Romanian Ministry of Funding. These groups included individuals from rural areas, economically disadvantaged individuals, people with disabilities, those without health insurance, unhoused individuals, members of the Roma community, persons lacking identity documents, single-parent families, individuals with addictions to alcohol, drugs, or other substances, as well as victims of domestic violence and human trafficking. These categories were selected based on socio-economic disadvantages, limited access to healthcare, and an increased risk of exposure to factors associated with HBV infection.

Over the designated period, 320,000 patients, each assigned a unique identification code, were included in the screening program. These participants resided in 24 counties located in the southern and eastern regions of the country, covering 58.5% of Romania’s 41 counties. Notably, Bucharest, the capital city situated in the southern part of Romania, was excluded from the program, as its population was deemed to have significantly better access to high-quality medical services compared to the targeted counties. A majority of the screened individuals, specifically 230,310 people (71.97%), lived in rural areas, while the remaining 89,690 people (28.03%) were urban residents. More than 1500 healthcare professionals, including family physicians (FP) and specialists in gastroenterology/hepatology, as well as nurses working in public medical institutions or under contracts with the Health Insurance Administration, took part in the screening conducted across the North-East, South-East, South, and South-West regions of the country. Rapid diagnostic tests (RDT) were performed by FP. A questionnaire was used to collect information on the sociodemographic characteristics (age, sex, geographic region of residence, and rural/urban status) of the participants and the potential HBV/HDV transmission risk factors. The face-to-face interviews were conducted by the FP at the same time as the RDT collection.

The healthcare facilities serving as staging and testing centers for patients identified as HBV-positive through preliminary rapid diagnostic tests were as follows: in Bucharest—the Fundeni Clinical Institute and the Bucharest University Emergency Hospital; in Craiova—the County Emergency Clinical Hospital; in Iași—the “Sf. Spiridon” County Emergency Clinical Hospital; and in Constanța—the “Sf. Apostol” County Emergency Clinical Hospital. Written informed consent was obtained from all participants prior to their enrolment. The study included patients who tested positive for hepatitis B surface antigen (HBsAg) during routine screenings conducted across multiple tertiary referral centers in Romania. The study population comprised 6815 HBsAg-positive individuals. All participants underwent systematic HDV screening, including testing for HDV antibodies (HDV Ab) and reflex HDV RNA testing for those with positive HDV Ab results. Clinical data were collected at the hepatology clinic visit: fibrosis staging, liver stiffness measurements (via transient elastography), abdominal ultrasound, serologic markers of HBV (HBeAg, antiHBeAb, antiHBsAb, and HBV DNA), HDVAb, and HDV RNA levels (ELISA kits and real-time PCR-based kit, Bosphore HBV Quantification Kit, Anatolia Geneworks, Istanbul, Turkey). The study was approved by the Local Ethics Committee (No. 8537/11 August 2020) and adhered to the principles of the Declaration of Helsinki.

### 2.2. Statistical Analysis

The crude prevalence of HBV and HDV chronic infection was calculated as proportions with 95% confidence intervals (CIs). Univariate comparisons between categorical variables were performed using the Chi-square test. Differences in continuous variables were evaluated using the Wilcoxon rank-sum test for non-parametric data. The trend analysis for ordered variables, such as age groups or fibrosis stages, was performed using a Wilcoxon-type test to assess statistical significance. All statistical tests were two-sided and a value of *p* less than 0.05 was used to indicate statistical significance. All statistical analyses were carried out using STATA/SE 11 software (StataCorp, College Station, TX, USA).

## 3. Results

### 3.1. Demographic and Epidemiological Data

These data regarding the whole study population were already published in our recent article about HBV epidemiology [2]. The risk factors identified for HBV infection are shared with HDV co-infection.

### 3.2. Study Population and Prevalence of HBV/HDV Co-Infection

A total of 6813 patients who tested positive for HBsAg were included in the study. Among these, 4.87% were positive for HDV antibodies (HDV Ab), indicating a significant burden of HDV exposure. Of the HDV-positive (HDV Ab) individuals, 75.6% demonstrated detectable HDV RNA, confirming active HDV replication. When dividing into LIVERO2 SOUTH and EST, the overall HDV prevalence was higher in the EST (5.82%) vs. 3.99% in the Southern Region of Romania. The prevalence of HDV RNA positivity was notably higher in the Eastern region (80.56%) compared to the Southern region (66.67%; *p* = 0.0001). The HDV co-infection was more prevalent among men compared to women (5.41% vs. 4.55%). Also, co-infection was more prevalent in the rural part (58.74%) of Romania versus urban parts (41.26%, *p* = 0.00006), so more prevalent among vulnerable people. The majority of co-infected patients were aged 30–49 years and another peak of prevalence was between 60 and 69 years. Patients with HBV/HDV co-infection were significantly younger than those with HBV mono-infection (mean age 49.28 years vs. 56.82 years; *p* = 0.0001). Table 1 presents the distribution of age groups among patients with and without HDV infection, providing a detailed breakdown of the proportion of HDV-positive individuals within each age group and highlighting statistically significant differences. HDV-positive individuals are more likely to be associated with vulnerable groups, as evidenced by the higher proportions of HDV positivity (10.55% vs. 7.69%, *p* < 0.0001). In addition, HDV-positive individuals are disproportionately concentrated in the lowest educational levels (ISCED 0-1, no education, or primary education category, accounting for 42.1% of positive cases). There is a clear decline in HDV positivity as educational attainment increases (*p* < 0.0001). The analysis of the data also reveals a significant association between marital status and HDV antibody positivity (*p* < 0.0001). Individuals who are never married show a disproportionately high rate of HDV positivity (21.52%), suggesting that this group may face higher exposure to risk factors for HDV, such as unsafe behaviors or healthcare disparities, while widowed and divorced/separated individuals have lower HDV positivity rates.

### 3.3. Clinical Characteristics of HBV and HBV/HDV Co-Infection

#### 3.3.1. Liver Stiffness and Fibrosis Staging

The mean liver stiffness measurement (LSM) via FibroScan was significantly higher in patients with HDV compared to those without HDV co-infection (12.76 kPa vs. 7.43 kPa, *p* = 0.0001). HDV-positive patients had a wider range of LSM values (minimum: 3 kPa, maximum: 75 kPa) compared to HDV-negative individuals (minimum: 1.5 kPa, maximum: 55 kPa). A detailed distribution of fibrosis stages revealed a significant increase in advanced fibrosis (F4) among HDV-positive patients (30.6%) compared to HDV-negative patients (9.7%, *p* = 0.0001). Conversely, early fibrosis stages (F0, F1) were more prevalent in HDV-negative individuals (49.3% and 18.2%, respectively) (see Figure 1).

#### 3.3.2. Controlled Attenuation Parameter (CAP) and BMI

CAP values, indicative of liver steatosis, were significantly lower in HDV-positive patients (222.25 ± 65.05 dB/m) than in HDV-negative individuals (250.42 ± 63.83 dB/m, *p* = 0.0001). Body mass index (BMI) was also notably lower in HDV-positive patients (25.92 ± 5.03 kg/m^2^) compared to those without HDV (27.44 ± 4.13 kg/m^2^, *p* = 0.0001), concordant with the presence of steatosis. There is a significant association between advancing fibrosis stages and increasing steatosis levels, as indicated by CAP values (*p* < 0.0001). In the early stages of fibrosis (F0 and F1), most patients have lower CAP values (<280 dB/m), corresponding to steatosis grades 0–2. As fibrosis progresses (F2 and F3), there is a marked increase in the proportion of patients with higher CAP values (≥280 dB/m), indicative of steatosis grade 3. Although a slight decrease in this trend is observed at stage F4, the overall pattern clearly demonstrates that steatosis severity tends to increase with fibrosis progression.

#### 3.3.3. HBV DNA Levels by HDV Antibody and RNA Status

A strong inverse correlation was observed between HDV positivity and HBV DNA levels: HDV-positive patients had a significantly higher proportion of HBV DNA < 2000 IU/mL compared to those HDV negative (*p* < 0.0001) (see Figure 2). HDV-positive patients were less likely to have HBV DNA > 2000 IU/mL, indicating the suppressive effect of HDV on HBV replication. In HDV-positive patients, HDV RNA positivity did not show a statistically significant association with HBV DNA levels (*p* = 0.66), although similar trends were noted. Among patients with active HDV replication, the median HDV viral load was 10,297.4 IU/mL, with a wide range of detectable levels up to 3,572,987.2 IU/mL.

### 3.4. Associated Comorbidities

Psychiatric comorbidities were significantly more prevalent in HDV-positive patients (11.03%) compared to HDV-negative individuals (3.89%, *p* = 0.001). Arterial hypertension was notably higher in patients with positive HDV (5.76%) compared to those without HDV co-infection (4.92%, *p* = 0.001). Diabetes mellitus prevalence was slightly higher in HDV-negative patients (9.08%) compared to HDV-positive individuals (3.55%, *p* = 0.110), though the difference was not statistically significant. The same was observed for thyroid disorders (2.89% in HDV positive vs 4.99% in HDV negative, *p* = 0.06). The prevalence of oncological disorders did not differ significantly between the two groups (4.14% in HDV positive vs. 4.89% in HDV-negative patients, *p* = 0.855).

## 4. Discussions

This study highlights the extent of HDV infection and its clinical implications in Romania while emphasizing its association with evolving demographic and socio-economic vulnerabilities. Building on our previous population-based survey (2015) [1] and hospital-based study (2024) [5], this analysis introduces a new focus on specific vulnerable populations, offering a more nuanced understanding of HDV epidemiology and its intersection with social determinants of health. It confirms the substantial burden of HDV and underscores the urgent need for targeted public health interventions.

### 4.1. Epidemiological Insights

HDV co-infection was detected in 4.87% of HBV-positive patients, a prevalence consistent with previous reports from Eastern Europe [3]. This prevalence is lower than the 23.1% HDV positivity reported in the 2015 population-based survey [1] but aligns with the 33.1% hospital-based prevalence found in the 2024 study, which focused on high-risk patients in tertiary referral centers [5]. The findings align with recent reports highlighting HDV prevalence in Eastern Europe, such as a Greek study on HDV and HIV co-infection that underscores regional disparities and their public health significance [12]. The lower prevalence here may reflect differences in study design and broader inclusion criteria, targeting a more diverse population. These results are consistent with studies that report varying prevalence rates of HDV among HBV-infected populations, indicating a substantial burden of HDV exposure in our cohort [13,14,15]. Notably, 75.6% of the HDV-positive individuals demonstrated detectable HDV RNA, confirming active viral replication. This aligns with the understanding that HDV co-infection exacerbates liver disease [16,17]. Geographical distribution showed significant differences between Eastern and Southern Romania, with prevalence rates of 5.82% and 3.99%, respectively. This disparity may reflect regional variations in risk factors, healthcare access, and economic conditions, as suggested by other epidemiological studies [18,19]. Furthermore, higher HDV RNA positivity in the Eastern region (80.56%) compared to the Southern region (66.67%) highlights the need for targeted public health interventions in high-risk areas [20,21]. Higher prevalence was observed in rural areas (58.74% vs. 41.26% in urban settings; *p* = 0.00006) and among younger age groups (30–49 years), with a secondary peak at 60–69 years. These findings align with known geographic and cohort effects. Recent findings published by Wranke et al. [22] reinforce these patterns, highlighting similar age-specific prevalence trends in HDV infection, particularly in rural and underserved populations across Eastern Europe and Central Asia. The increased exposure to risk factors in younger, economically active populations and historical exposure in older cohorts prior to HBV vaccination programs likely explain this distribution [23,24]. This observation aligns with previous studies that identified rural residency, economic migration, and limited healthcare access as significant risk factors for HBV and HDV infection in Eastern European countries, pointing to systemic inequities that exacerbate the disease burden in vulnerable populations. Furthermore, cultural practices and healthcare disparities in Eastern Europe may contribute to distinct epidemiological profiles, necessitating tailored interventions to reduce transmission and improve outcomes.

HDV-positive individuals were significantly younger than HBV mono-infected patients, consistent with the aggressive nature of HDV infection [25]. Educational attainment and marital status emerged as significant predictors of HDV positivity. Individuals with no formal education or only primary education (42.1%) were disproportionately represented among HDV-positive cases, aligning with prior research linking low education levels to higher infectious disease risk [26]. Similarly, never-married individuals had the highest HDV positivity rate (21.52%), possibly reflecting greater exposure to high-risk behaviors [27]. Our data emphasize the emergence of distinct vulnerable populations, including those with lower socioeconomic status and limited education. This finding is consistent with literature indicating that disadvantaged groups are at greater risk for HBV and HDV infections [28,29]. Findings from a recent review on global HDV prevalence, which reported significant socioeconomic disparities in HDV infection rates, further corroborate these results [30]. These results reaffirm that HDV remains a disease of inequity, disproportionately affecting vulnerable populations and highlighting an evolving epidemiological profile in Romania.

### 4.2. Clinical Features and Disease Burden

The clinical severity of HBV/HDV co-infection is evident in the higher liver stiffness measurements (LSM) observed in HDV-positive patients. Liver cirrhosis was significantly more prevalent among HDV-positive patients, highlighting HDV’s role in accelerating liver damage. Conversely, lower fibrosis stages (F0 and F1) were predominantly seen in HDV-negative individuals. This supports previously published studies that associate HDV with a higher risk of decompensated liver cirrhosis, HCC, and the need for liver transplantation [31,32,33].

CAP values and BMI were lower in HDV-positive individuals, reflecting reduced steatosis in this group. This inverse relationship between HDV and steatosis may result from metabolic suppression and inflammation-driven pathophysiology, as observed in other studies [34]. The association between fibrosis progression and increasing steatosis observed in CAP measurements reinforces the importance of metabolic monitoring in these patients, as well as the need for comprehensive management strategies that address both viral replication and liver health in co-infected patients [35,36]. A study analyzing chronic hepatitis B patients found that those co-infected with HDV had significantly lower rates of hepatic steatosis. Specifically, the cumulative probability of HCC was 2.88% in patients without steatosis, 1.56% in those with mild-to-moderate steatosis, and 0.71% in those with severe steatosis, suggesting an inverse relationship between steatosis severity and HCC risk in the context of HBV infection [37,38]. Additionally, HDV co-infection was associated with lower HBV DNA levels, corroborating HDV’s suppressive effect on HBV replication. However, no significant association was found between HDV RNA and HBV DNA levels, indicating complex host-virus interactions [39]. Notably, all HDV infections in Romania belong to genotype 1 [40,41]. This genotype dominance underscores the homogeneity of HDV in the region, making genotyping unnecessary for clinical decision-making. With the introduction of bulevertide therapy, which is effective regardless of genotype [42], the focus on genotype-specific interventions has diminished.

### 4.3. Comorbidities

The prevalence of associated comorbidities was also examined, revealing that psychiatric comorbidities were significantly more prevalent in HDV-positive patients (11.03%) compared to their HDV-negative counterparts (3.89%). This may reflect the psychological burden of severe liver disease or socio-economic challenges faced by vulnerable groups. Integrated care approaches addressing both physical and mental health are essential for HBV/HDV co-infected patients.

Additionally, arterial hypertension was more common in HDV-positive patients, but diabetes mellitus, thyroid disorders, and oncological conditions did not show significant differences between the two groups. This suggests that HDV’s impact on comorbidities may be selective or related to age. Our findings highlight the intricate relationship between HBV and HDV, particularly concerning their epidemiology, clinical characteristics, and associated comorbidities.

This study underscores a critical shift in HDV epidemiology—the identification of vulnerable populations as a distinct group requiring targeted intervention. While the population-based survey (2015) [1] provided a baseline understanding of HDV prevalence across Romania, and the hospital-based study (2024) [5] highlighted the clinical impact in tertiary care settings, this analysis bridges these perspectives by focusing on the socioeconomically disadvantaged, rural, and undereducated population.

Our results show the importance of implementing systematic and double reflex testing in all HBsAg-positive individuals. These measures are essential to address the high burden of HBV/HDV co-infection, reduce transmission, and improve patient outcomes.

Our study has several limitations that should be considered. The screening program targeted predefined vulnerable populations in Romania, as outlined by the European Social Fund criteria. Other potentially vulnerable groups, such as individuals living with HIV, men who have sex with men (MSM), survivors of sexual assault, individuals in conflict zones or refugees, older adults in long-term care facilities, and migrants or sex workers, were not included. This may limit the generalizability of the findings to all at-risk groups. However, the vulnerable populations defined by the Romanian Ministry of Funding reflect those most at risk for viral infections in Romania. Additionally, Bucharest was excluded from the program due to its better healthcare access, potentially underrepresenting urban populations with healthcare disparities. Sociodemographic and risk factor data were self-reported, which may introduce recall bias or inaccuracies, and certain behavioral or environmental risk factors were not included, limiting the comprehensiveness of the analysis. Lastly, while a higher prevalence of psychiatric comorbidities was observed in HDV-positive patients, causal relationships and their impact on outcomes were not explored.

The findings emphasize the urgent need for tailored strategies to address HDV infection in Romania. These include targeted screening and education initiatives, particularly for rural areas and socially or economically disadvantaged populations; expanded vaccination and treatment access, with a focus on improving HBV vaccination coverage among vulnerable groups and ensuring novel treatments like bulevirtide are available to all HDV RNA-positive patients; and integrated care models that incorporate management of psychiatric and cardiovascular comorbidities to improve outcomes for affected individuals. The significant burden of HBV and HDV co-infection in Romania highlights a critical public health concern, compounded by socio-economic and healthcare challenges. Enhanced screening, vaccination, and education efforts are vital to reducing the impact of these infections. Future research should focus on longitudinal studies to understand the long-term outcomes of HBV/HDV co-infection and evaluate the effectiveness of new treatment strategies while addressing the underlying factors that perpetuate high co-infection rates. These efforts are essential to develop evidence-based public health policies tailored to Romania’s unique context.

## Figures and Tables

**Figure 1 viruses-17-00052-f001:**
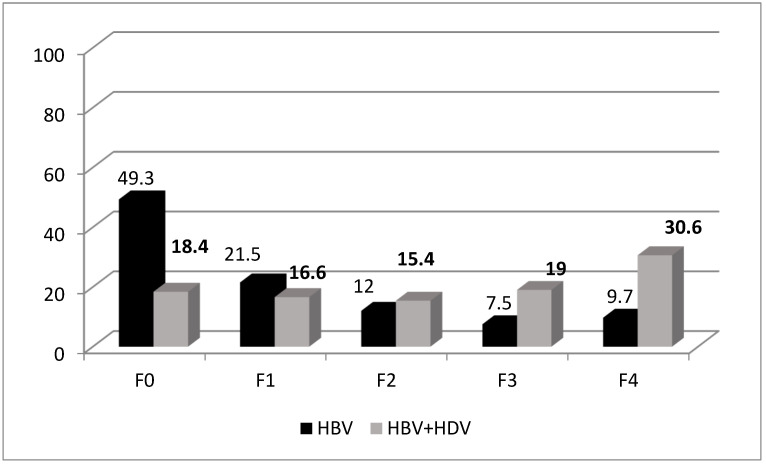
Fibrosis stages distribution according to the presence of HDV co-infection.

**Figure 2 viruses-17-00052-f002:**
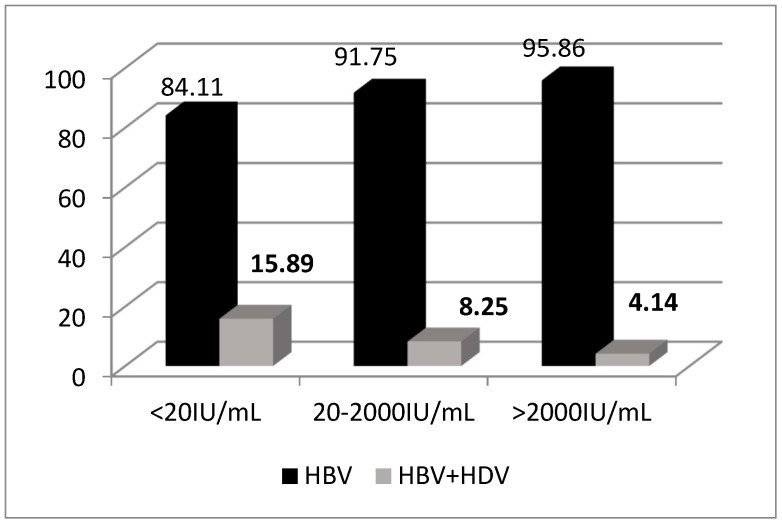
Quantitative HBV DNA according to HDV positivity.

**Table 1 viruses-17-00052-t001:** Age group distribution by HDV status.

Age Groups by HDV Positivity						
	HDV Negative		HDV Positive		Total	
Age Group (Years)	N	%	N	%	N	%
18–29	101	1.56	3	0.9	104	1.53
30–39	921	14.21	112	33.73	1033	15.16
40–49	1028	15.86	68	20.48	1096	16.09
50–59	1391	21.46	50	15.06	1441	21.15
60–69	1720	26.54	74	22.29	1794	26.33
70–79	1048	16.17	23	6.93	1071	15.72
80–89	268	4.14	2	0.6	270	3.96
90–103	4	0.06	0	0	4	0.06
All VHB patients	6481	95.13	332	4.87	6813	100
Chi test *p* = 0.0001						

## Data Availability

https://screening-insp.ro/activitati-derulate/comunicate-presa (accessed on 1 July 2024).

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
