# Peer review of "Unveiling Prevalence, Risk Factors, and Outcomes of Hepatitis D Among Vulnerable Communities in Romania"

_viruses, 2024, doi:10.3390/v17010052_

Round 1

Reviewer 1 Report

Comments and Suggestions for Authors

Thank you for the opportunity to review this manuscript. This is a comprehensive study investigating the epidemiology, risk factors, and clinical outcomes of HBV/HDV co-infection in vulnerable populations in Romania. This is a very interesting report, however I have some suggestions/concern

1.      How is this “vulnerable” population chosen ? Why are other vulnerable groups excluded ? HIV, immigrants, sex workers. The title is misleading. These populations should be added, or title changed or discussed about, including the limitation section

2.      A stand alone limitation section should be added. Several limitations are identified

3.       HDV-positive individuals, means Ab positive ? nucleic acid positive ? Please define and keep terminology uniform

4.      Please add references in your statements referring to data e.g line 59 and all along the way of text

5.      Important studies are missing including regional data from Eastern Europe and vulnerable populations e.g HIV etc

i.e. Int J Infect Dis. 2001;5(4):209-13, J Gastrointestin Liver Dis. 2010 Mar;19(1):43-8. ;  Viruses  2024 Jun 28;16(7):1044. Liver Int. 2024 Sep;44(9):2442-2457,  J Viral Hepat. 2024 Feb;31(2):120-128.

Further in depth discussion should be performed in alignment with global literature

6.      Table of demographics and comparisons and statistic analysis is missing. It is pivotal besides figures and reference [2], for this 4.87%

Author Response

Reviewer 1

Comment 1: How is this “vulnerable” population chosen? Why are other vulnerable groups excluded, such as individuals living with HIV, immigrants, and sex workers? The title is misleading. These populations should be added, or the title changed, or their exclusion discussed in the limitations section.

Response: Thank you for your valuable comment. The specific populations considered vulnerable in Romania were chosen based on criteria established by the European Social Fund and adapted by the Romanian Government. These criteria were specifically outlined in the POCU Project, which was financed by the European Social Fund. The project required our Ministry of Funding to focus screening efforts exclusively on certain predefined vulnerable groups. Vulnerability in this context was defined by the Ministry and included the following categories:

  • People from rural areas
  • Individuals living in poverty
  • People with disabilities
  • Uninsured individuals
  • Unhoused people
  • People of Roma ethnicity
  • Individuals without identity documents
  • Single-parent families
  • Individuals suffering from addiction to alcohol, drugs (including intravenous drug users, IVDUs), and other toxic substances
  • Victims of domestic violence and human trafficking

These groups were prioritized due to their socio-economic disadvantages, limited access to healthcare, and increased exposure to risk factors associated with HBV infection.

Other populations, such as individuals living with HIV, were not included in this specific project because they are already systematically screened and managed for viral liver infections by infectious disease specialists as part of national health protocols. Similarly, while immigrants and sex workers may be considered vulnerable in other contexts, they were not part of the target population defined by this project’s scope.

We believe the title accurately reflects the populations included in our study and their vulnerability, as defined by the project guidelines. However, to address your concern, we have clarified these points in the Materials and Methods section, emphasizing the rationale for selecting these groups and the exclusion of others. Additionally, we have acknowledged these exclusions as a limitation in the Discussion section.

Comment 2: A stand alone limitation section should be added. Several limitations are identified.

Response: Our study has several limitations, which we acknowledge and have integrated into the Discussion section (see the Discussion part highlighted in yellow) of the article:

  1. Population Scope: The screening program targeted predefined vulnerable populations in Romania, as determined by the European Social Fund criteria. Other potentially vulnerable groups, such as individuals living with HIV, men who have sex with men (MSM), survivors of sexual assault, individuals in conflict zones or refugees, older adults in long-term care facilities, and migrant or sex workers, were not included. This limitation may affect the generalizability of findings to all at-risk groups. However, the chosen vulnerable populations reflect those most prevalent and at risk for viral infections in Romania, as defined by the Romanian Ministry of Funding.
  2. Geographical Exclusion: Bucharest, the capital city, was excluded from the program due to its superior healthcare access compared to other regions. While this aligns with the project's focus on underserved areas, it may have underrepresented urban populations with healthcare disparities elsewhere.
  3. Self-Reported Data: Sociodemographic and risk factor data were collected via self-reported questionnaires, which may be subject to recall bias or inaccuracies, potentially impacting the robustness of reported associations.
  4. Unmeasured Confounders: Certain behavioral or environmental risk factors were not captured, which may have influenced the observed associations. Future studies should aim to include these factors for a more comprehensive analysis.
  5. Psychiatric Comorbidities: The study observed a higher prevalence of psychiatric comorbidities among HDV-positive patients, highlighting the need for integrated care approaches. However, the study did not explore causal relationships or the impact of these comorbidities on patient outcomes.

Despite these limitations, the study offers critical insights into the HBV/HDV burden among vulnerable populations in Romania, providing a foundation for targeted public health interventions and guiding future research.

Comment 3: HDV-positive individuals, means Ab positive? nucleic acid positive ? Please define and keep terminology uniform

Response: HDV-positive individuals means Ab positive; we have revised the manuscript and kept the terminology uniformly (see all the sentences marked in yellow).

Comment 4: Please add references in your statements referring to data e.g line 59 and all along the way of text

Response:       Thank you for your comment. However, from line 59 in our article, we have the Material and Methods section. We would like to clarify that this section provides a detailed description of the specific methodology and procedures applied in our project. The content reflects the factual realities of the study, including participant demographics, geographic coverage, healthcare professionals involved, and the diagnostic tools used. These details are intrinsic to our project and do not reference or rely upon previously published studies or external methodologies.

As such, there is no indication to include references in this section, as the information pertains solely to the operational aspects of our screening program and data collection process. Nonetheless, we have ensured that any relevant ethical approvals (e.g., the Local Ethics Committee approval) and adherence to established principles (e.g., the Declaration of Helsinki) are explicitly mentioned.

We trust this addresses your concern and ensures the clarity and accuracy of the manuscript.

Comment 5: Important studies are missing including regional data from Eastern Europe and vulnerable populations e.g HIV etc

i.e. Int J Infect Dis. 2001;5(4):209-13, J Gastrointestin Liver Dis. 2010 Mar;19(1):43-8. ;  Viruses  2024 Jun 28;16(7):1044. Liver Int. 2024 Sep;44(9):2442-2457,  J Viral Hepat. 2024 Feb;31(2):120-128.

Further in depth discussion should be performed in alignment with global literature

Response: Thank you for your comments. We have carefully reviewed the suggested references and points for discussion.

  1. HIV and Vulnerable Populations: While we acknowledge the importance of studying hepatitis D in people living with HIV and other vulnerable populations, our study did not included individuals co-infected with HIV. This exclusion was intentional, as the focus of our research was on the broader population-level epidemiology of HDV, which differs from studies targeting vulnerable subgroups. Nonetheless, we recognize the accelerated fibrosis progression in HIV/HDV co-infected individuals and will include a brief mention of this in the discussion to provide a more comprehensive context.
  2. Older Studies: The referenced articles from 2001 and 2010, while historically significant, do not reflect the current epidemiological trends, diagnostic advances, or therapeutic developments in HDV management. However, we will incorporate insights from the more recent references provided (e.g., Viruses, 2024; Liver International, 2024; J Viral Hepat, 2024) into the discussion, as they align with the current landscape of HDV research.
  3. Further Discussion and Global Literature Alignment: We agree that situating our findings within the global context is essential for a thorough analysis. We will expand the discussion to address regional differences in HDV epidemiology and the implications of new therapeutic approaches, referencing recent global and regional studies as appropriate.

We hope this clarification addresses your concerns, and we provided a revised discussion section that reflects these updates.

Comment 6: Table of demographics and comparisons and statistical analysis is missing. It is pivotal besides figures and reference [2], for this 4.87%.

Response: Thank you for your comment and suggestion. To enhance clarity and align with standard reporting practices, we have included in the revised manuscript a table summarizing the age distribution by HDV status and associated statistical analyses (see Table 1). We believe this addition addresses your request, and no further changes are necessary, as the manuscript already comprehensively covers the demographic and statistical analyses relevant to the study.

Reviewer 2 Report

Comments and Suggestions for Authors

This manuscript presents evidence concerning the prevalence of HBV and HDV in various populations in Romania.  The authors than correlate the prevalence with various factors including geography, ethnic background, wealth, education, etc.  The results are straight forward and are interesting in terms of their usefulness in planning public health interventions.  Since the authors determined the viral load of HDV by RNA analysis I think that the manuscript could be improved if the RNA analysis included sequencing to determine whether different variants of HDV were circulating in different populations. 

Author Response

Reviewer 2

Comment 1: This manuscript presents evidence concerning the prevalence of HBV and HDV in various populations in Romania.  The authors than correlate the prevalence with various factors including geography, ethnic background, wealth, education, etc.  The results are straight forward and are interesting in terms of their usefulness in planning public health interventions.  Since the authors determined the viral load of HDV by RNA analysis I think that the manuscript could be improved if the RNA analysis included sequencing to determine whether different variants of HDV were circulating in different populations. 

Response: Thank you for your suggestion. In Romania, all HDV infections have been shown to involve genotype 1, as confirmed by molecular epidemiology studies (e.g., Grecu et al., Pathogens 2024; Ricco et al., J Viral Hepat 2018). Genotype 1 is highly prevalent and uniform across the population in our region, making sequencing to determine genotype variation unnecessary in this context. Furthermore, in the era of bulevirtide therapy, HDV genotyping is not clinically relevant, as current treatment strategies are genotype-independent. Given this, we believe that the inclusion of HDV sequencing data would not significantly enhance the manuscript or contribute additional actionable insights. We incorporated this into the discussion and references part (marked in yellow).